# lillies: An R package for the estimation of excess Life Years Lost among patients with a given disease or condition

**Oleguer Plana-Ripoll**[1]*, **Vladimir Canudas-Romo**[2], **Nanna Weye**[1], **Thomas M. Laursen**[1], **John J. McGrath**[1,3,4], **Per Kragh Andersen**[5]

1 National Centre for Register-based Research, Aarhus University, Aarhus, Denmark, 2 School of Demography, ANU College of Arts & Social Sciences, Australian National University, Canberra, Australia, 3 Queensland Brain Institute, University of Queensland, St Lucia, Queensland, Australia, 4 Queensland Centre for Mental Health Research, The Park Centre for Mental Health, Wacol, Queensland, Australia, 5 Section of Biostatistics, University of Copenhagen, Copenhagen, Denmark

* opr@econ.au.dk

**Data Availability Statement:** All relevant code to replicate the results in the paper is available in the supplement, while all relevant data are available through the 'lillies' R package, freely available

## Abstract

Life expectancy at a given age is a summary measure of mortality rates present in a population (estimated as the area under the survival curve), and represents the average number of years an individual at that age is expected to live if current age-specific mortality rates apply now and in the future. A complementary metric is the number of Life Years Lost, which is used to measure the reduction in life expectancy for a specific group of persons, for example those diagnosed with a specific disease or condition (e.g. smoking). However, calculation of life expectancy among those with a specific disease is not straightforward for diseases that are not present at birth, and previous studies have considered a fixed age at onset of the disease, e.g. at age 15 or 20 years. In this paper, we present the R package lillies (freely available through the Comprehensive R Archive Network; CRAN) to guide the reader on how to implement a recently-introduced method to estimate excess Life Years Lost associated with a disease or condition that overcomes these limitations. In addition, we show how to decompose the total number of Life Years Lost into specific causes of death through a competing risks model, and how to calculate confidence intervals for the estimates using non-parametric bootstrap. We provide a description on how to use the method when the researcher has access to individual-level data (e.g. electronic healthcare and mortality records) and when only aggregated-level data are available.

## Introduction

Life expectancy at birth for a given population is defined as the life table average age at death. However, demographers are usually interested in estimating life expectancy for a population that is not yet extinct, and for which age at death is not available for everyone. For example, one could be interested in estimating life expectancy for babies born in year 2017. In such situations, life expectancy at birth is a summary measure of life tables based on mortality for that

through the Comprehensive R Archive Network (CRAN).

**Funding:** This work was supported by the European Union's Horizon 2020 research and innovation programme (Marie Sklodowska-Curie grant agreement No 837180 to Oleguer Plana-Ripoll), the Danish National Research Foundation (Niels Bohr Professorship to John J McGrath), and the National Health and Medical Research Council (John Cade Fellowship to John J McGrath). The funders had no role in study design, data collection and analysis, decision to publish, or preparation of the manuscript.

**Competing interests:** The authors have declared that no competing interests exist.

given year (estimated as the area under the survival curve built with all age-specific mortality rates in that specific year). This measure can be interpreted as the average number of years a newborn is expected to live if all age-specific mortality rates in that year remain constant in the future. In addition, it is possible to estimate life expectancy for different subgroups in the population (e.g. boys and girls), by considering the age-specific mortality rates within these specific subgroups, and assuming that a boy (girl) will experience current mortality rates for males (females) throughout their life. Global life expectancy at birth in year 2017 was 75.6 years for females and 70.5 years for males [1]. Females were therefore expected to live five years longer than males, or – put it in a different way – males were expected to lose five years of life compared to females.

The number of years of life lost can also be used to estimate how many years patients with a given disease are expected to lose compared to the general population, by subtracting the life expectancy in those with the disorder or condition of interest from that of the general population. Such measure is useful to quantify – and therefore compare – the societal burden of different diseases. Although this health metric is of interest for epidemiologists due to its potential impact, the estimation of life expectancy among those with a given disease is not straightforward (different methodologies have been described and discussed in detail elsewhere [2,3]). For congenital diseases (i.e. diseases that are present at birth), life expectancy among the diseased can be estimated using age-specific mortality rates among those with the disease, as we assume that they will experience these mortality rates throughout their entire life. However, for other disorders not present at birth, age of onset can vary widely between individuals. If the same approach is used for these types of disorders, the estimated life expectancy can only be interpreted as the expected lifetime for persons who have had the disease since birth, since there is again the assumption that the diseased experienced these mortality rates throughout their entire life – even before disease onset. In an attempt to overcome this limitation when calculating life expectancy for persons with a given disease, some studies have assumed that the persons with the disease experienced the mortality rates of the general population until a specific age threshold, and the specific mortality rates of the diseased afterwards (e.g. 15 years for mental disorders [4–6], 20 years for type 1 diabetes [7,8], and 55 years for colon cancer [9]). However, this simplifying assumption does not reflect the underlying age of onset distribution, and can result in biased estimates of life expectancy. Another approach has been to consider those diagnosed before a range of different possible ages [10,11], which then leads to a difficult interpretation and reporting of results given that there are different estimates for each age of onset anchor points.

Recently, new methods have been developed that overcome these past limitations. The Life Years Lost method [2,12] can estimate remaining life expectancy among those with a disorder of interest at the specific observed time of onset, and compare the average of these individuals to that of the general population of same age. In addition, one of the main advantages of this method is that it is possible to decompose the total life lost associated with a given disease into specific causes of death by means of a competing risks model [13], which permits the inspection of how specific causes of death contribute to the premature mortality in the disorder or exposed group of interest (e.g. smoking or any other time-varying or constant risk factor). Differences in remaining life expectancy after disease onset is relatively easy to interpret, and complements other widely used mortality estimates (e.g. standardized mortality rates or mortality rate ratios). This method has been used to investigate excess mortality associated with mental disorders [12,14,15]; the Life Years Lost measure found smaller differences in life expectancy compared to previous estimates that assumed a fixed age at onset at age 15 years [4–6].

In this paper, we present the R package `lillies` (a word that reflects the initials of Life Years Lost; LYLs), whose version 0.2.5 is available through the Comprehensive R Archive Network (CRAN) at http://CRAN.R-project.org/package=lillies, and will work with R version 3.5.0 or higher. We provide a description of how to estimate excess LYLs associated with a given disease or condition, including the decomposition into specific causes of death and the calculation of confidence intervals using bootstrapping. Additionally, we provide a description on how to use the method when the researcher has access to individual-level data (e.g. electronic healthcare and mortality records), but also when only aggregated-level data are available.

## Excess Life Years Lost among patients with a given disease: the method

The LYL method, which is based on a population of persons with a given disease, uses age at disease diagnosis for each person in the population as its starting point and estimates the expected residual lifetime at that age using age-specific mortality rates. The number of excess LYLs is estimated by matching the expected residual lifetime for someone diagnosed with the disease, with the life expectancy from the general population at that specific age. Interestingly, age-specific life expectancy in the general population is usually available through standard life tables from the Central Bureau of Statistics in each country. By using life tables from the general population, a metric can be calculated that compares those with a disorder of interest to a group of persons from the same source population matched on age and sex. In order to obtain an overall single estimate of excess LYL (instead of one estimate for each affected person), it is possible to take an average of all the person-specific LYL. This estimate can be interpreted as the average life lost (in years) that patients with a given disease experience from the time of diagnosis *in excess* to those experienced by a reference population of same age.

## Excess Life Years Lost using individual-level data

### Step 1: Selection of the study population

A simulated dataset is used to show how to use the LYL method (the R code is available in S3 Appendix). The 'simu_data' (available through the `lillies` package), contains information on a simulated population of 100,000 persons. In this population, all individuals are followed from birth (age_start=0 for everyone) until death (at age_death, which ranges from 0.25 until 95 years). Note that persons alive at age 95 years (n=4,384) are censored. While censoring at age 95 years is the only censoring mechanism applied in this simulated population, the LYL method also works with censoring at different ages, as in a standard time-to-event analysis (and therefore assuming independent censoring, i.e. those being censored at one specific time should be representative of those still at risk at that time). In addition, delayed entry (i.e. left truncation) is also possible in data sets where some individuals first enter the study some years after birth (in that case, variable `age_start` would contain the age at start of follow-up). Additionally, all deaths were classified into two mutually exclusive and collectively exhaustive causes: natural and unnatural causes. Among those who died before age 95 years, 89,989 (94.1%) died of natural causes and 5,627 (5.9%) of unnatural causes. Finally, 32,391 persons in this simulated population experienced a disease of interest, and the age of onset for these individuals (mean age 38.9 years) is recorded in the variable `age_disease`.

```
install.packages("lillies")

library(lillies)

data(simu_data)

summary(simu_data[, c("age_death", "death", "cause_death",
"age_disease")])
```

The R output is shown in Fig 1.

In order to estimate life expectancy among persons with a given disease, it is necessary to identify those who experience the disease. However, the entire population will be used in Step 3 to make the comparison between the two groups. In case the researcher has access to only a group of persons with a disease (instead of the entire population), we show (in Step 5) how to compare it to the general population using publicly available life tables.

```
diseased <- simu_data[!is.na(simu_data$age_disease), ]

nrow(diseased)

## [1] 32391
```

## Step 2: Life Years Lost at one specific age for persons with the disease (age 45 years as example)

The first step is to calculate remaining life expectancy for each person at time of diagnosis. The age at disease onset ranged from 1.25 to 94.75 years; it is therefore necessary to calculate remaining life expectancy at each age from 1 to 94 years (if considering only integers as possible ages of onset). For example, for someone diagnosed at age 45 years, the conditional survival function is shown in Fig 2A. This curve is built on mortality rates obtained from all persons in the population with the disease who were still alive at age 45 years (this restriction is similar to a Landmark analysis that conditions on those alive at specific landmark points [16,17]). As example, around 40% of individuals are still alive at age 75 years, while the remaining 60% have died before.

```
summary(simu_data[, c("age_death", "death", "cause_death", "age_disease")])
##    age_death        death            cause_death      age_disease
##  Min.   : 0.25   Mode :logical   Alive    : 4384   Min.   : 1.25
##  1st Qu.:68.25   FALSE:4384      Natural  :89989   1st Qu.:18.75
##  Median :78.50   TRUE :95616     Unnatural: 5627   Median :31.50
##  Mean   :75.55                                     Mean   :38.95
##  3rd Qu.:86.25                                     3rd Qu.:57.00
##  Max.   :95.00                                     Max.   :94.75
##                                                    NA's   :67609
```

**Fig 1. R Output 1.**

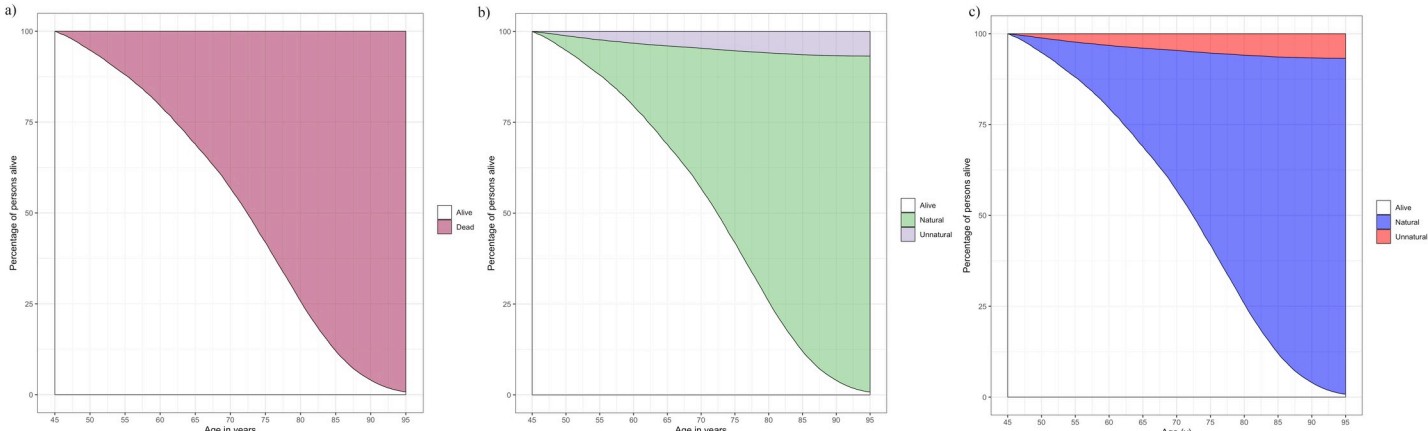

**Fig 2.** Conditional survival curves (a, b and c), stacked cumulative incidence for all-cause mortality (a) and stacked cause-specific cumulative incidences for natural and unnatural causes of deaths (b and c) for persons with a diagnosis of the disease and alive at age 45 years. Fig 2C is the same as Fig 2B but changing the colors and the x axis label. Details on how to interpret these figures are available in S1 Appendix.

The basic features of survival analysis that are required to implement this method are provided in S1 Appendix. In brief, remaining life expectancy at age 45 years is estimated as the area under the conditional survival curve from age 45 years to ∞; however, this measure is sometimes ill-determined if there are censored observations and the curve does not reach zero (i.e. some persons are still alive at the end of the curve), as it is the case in this example (in fact, the survival curve cannot reach zero if the last person at risk is censored, even if this is the only censored observation in the dataset). One approach to overcome this limitation is the *τ-restricted mean lifetime*, which can be interpreted as the average number of years lived before time *τ*, and is defined as the area under the curve until time *τ*: For this example, *τ* has been set to 95 years, an age in which persons were censored if they had not died before. The estimate of LYL has therefore to be interpreted as life lost after the specific age (45 years in this example) and before age 95 years. Although the choice of 95 years is arbitrary, the life lost before *τ* can be interpreted as total life lost if *τ* is an age in which the survival probability is as low as possible (ideally zero). However, in other settings, the researchers might be interested in LYL before age 18 years for childhood disorders, or before retirement at age 68 years, for example. The remaining 95-restricted life expectancy at age 45 years (referred to as $_{50}e_{45}$ in S1 Appendix), is therefore 26.1 years. A person with disease onset at age 45 years lives on average an additional 26.1 years before age 95 years; or – put it in a different way – persons with disease onset at age 45 years lose 23.9 years of life before reaching 95 years. Estimates at one specific age can be obtained with the function `lyl`, which later can be examined with functions `summary` and `plot`.

```
LYL45 <- lyl(data = diseased, t0 = age_disease,
t = age_death, status = death, age_specific = 45, tau = 95)

summary(LYL45)
```

The R output is shown in Fig 3.

Note that the beginning of follow-up here is at `age_disease` to avoid immortal time bias (individuals survive until disease onset, therefore follow-up must start at disease onset, and not at birth, for the group of persons with the disease). In addition, it is possible to

estimate LYL due to specific causes of death [12,13], e.g. natural and unnatural causes of death (by providing the variable `cause_death`, which is a categorical variable with different causes, instead of `death`). The total of 23.9 years of life that those diagnosed at age 45 years lose before age 95 years can be decomposed into 21.7 years due to natural causes and 2.2 years due to unnatural causes.

```
LYL45 <- lyl(data = diseased, t0 = age_disease,
t = age_death, status = cause_death, age_specific = 45,
tau = 95)

summary(LYL45)

plot(LYL45)
```

The R output is shown in Fig 4 and the plot generated is shown in Fig 2B.

Fig 2B shows the survival function for persons with disease onset at age 45 years, in an analogous way as in the figure with all-cause mortality (Fig 2A); however, the area above the survival curve (corresponding to the LYL) is now decomposed into natural and unnatural causes of death. In order to decompose the area, the cumulative incidence function for one of the causes has to be 'stacked' over the survival curve [13] (details on how to interpret these figures are available in S1 Appendix). Among the 60% who have died before age 75 years, around 50% were due to natural causes and 10% due to unnatural causes. Note that it is possible to change the colors of the figures by adding the `colors` option, or change other attributes using the standard `ggplot2` notation.

```
plot(LYL45, colors=c("blue", "red")) + ggplot2::xlab("Age
(y)")
```

The plot generated is shown in Fig 2C.

At this point, it is possible to estimate confidence intervals (CI) via non-parametric bootstrap with the function `lyl_ci` (confidence level – 0.95 by default – can be specified with the parameter `level`). The total excess LYL for those with the disease of interest is 23.9 (95% CI: 23.7 – 24.1).

```
LYL45_ci <- lyl_ci(LYL45, niter = 1000)

summary(LYL45_ci)
```

The R output is shown in Fig 5.

## Step 3: Comparison to the general population

On average, individuals with disease onset at age 45 years live an additional 26.1 years, which means they lose 23.9 years of life when considering a theoretical maximum of 95 years. It is now possible to estimate the life expectancy and life lost for the general population of same

```
summary(LYL45)

## Estimates at age 45 years [maximum age tau = 95 years]
##
## ========================  ========  =======  ========
## \                         estimate  CI_left  CI_right
## ========================  ========  =======  ========
## Remaining life expectancy    26.10       -         -
## Total Life Years Lost        23.90       -         -
## ========================  ========  =======  ========
## *Confidence intervals can be estimated with function 'lyl_ci'
```

**Fig 3. R Output 2.**

age. The difference between the two estimates results in the *excess* LYL, which is the amount of life persons with a disease at age 45 years lose *in excess* of that seen in the general population.

```
LYL45_ref <- lyl(data = simu_data, t = age_death,
status = cause_death, age_specific = 45, tau = 95)

summary(LYL45_ref)
```

The R output is shown in Fig 6.

On average, individuals in the general population alive at age 45 years live an additional 32.3 years, which means that a person with disease onset at age 45 years lose on average 6.2 years compared to a person of the same age from the general population (32.3 – 26.1 = 6.2). Or alternatively, a person with disease onset at age 45 years experiences an excess LYL of 6.2 years (23.9 vs. 17.7 years). Note that start of follow-up (parameter t0) is not specified because all persons are followed from birth (it should be specified in situations with delayed entry). The excess LYL among those with a disease compared to the general population at age 45 years of 6.2 years can be decomposed into 5.0 years due to natural causes and 1.2 years due to unnatural causes.

```
lyl_diff(LYL45_ci, LYL45_ref)
```

The R output is shown in Fig 7.

The function lyl_diff compares two objects, which can be provided with confidence intervals (as LYL45_ci) or without them (as LYL45_ref). When estimating confidence intervals for the difference, the object without confidence intervals is assumed to be estimated without uncertainty (this assumption might be reasonable if the entire population is available). Finally, it is possible to draw the two survival curves side by side with the function lyl_2plot.

```
lyl_2plot(LYL45, LYL45_ref)
```

```
summary(LYL45)

## Estimates at age 45 years [maximum age tau = 95 years]
##
## ========================  ========  =======  ========
## \                         estimate  CI_left  CI_right
## ========================  ========  =======  ========
## Remaining life expectancy    26.10        -         -
## Total Life Years Lost        23.90        -         -
## - Due to Natural             21.75        -         -
## - Due to Unnatural            2.15        -         -
## ========================  ========  =======  ========
## *Confidence intervals can be estimated with function 'lyl_ci'
```

**Fig 4. R Output 3.**

The plot generated is shown in Fig 8. On average, we can see that, for those alive at age 45 years, mortality after this age is much lower for a person from the general population than for a person with the disease of interest.

The main reason to compare those with a given disease to the general population – and not to persons without the disease – is that the number of LYL at a given age, e.g. 45 years, is estimated using mortality rates at ages 45 years and beyond. By choosing persons without the disease as a comparison group, we would assume that someone who has not experienced the disease at age 45, would remain free of the disease until death. Although it might seem problematic to include persons with a disease in both the diseased and reference groups, this is analogous to standardized mortality ratios, which compare mortality in a group of persons to the one in the general population [18]. In any case, differences in life expectancy would be even larger if the comparison group were persons without the disease.

### Step 4: Life Years Lost over a range of different ages

We have presented how to estimate LYLs at age 45 years. In order to obtain a summary measure of LYLs associated with the disorder of interest, remaining life expectancy after diagnosis for each person in the group of individuals with the disease has to be calculated. This is the same as estimating them for each specific age from 1 to 94 years, and then weight the average

```
summary(LYL45_ci)

## Estimates at age 45 years [maximum age tau = 95 years]
##
## ========================  ========  =======  ========
## \                         estimate  CI_left  CI_right
## ========================  ========  =======  ========
## Remaining life expectancy    26.10    25.94     26.25
## Total Life Years Lost        23.90    23.75     24.06
## - Due to Natural             21.75    21.60     21.91
## - Due to Unnatural            2.15     2.04      2.26
## ========================  ========  =======  ========
## *95% confidence intervals based on 1000 bootstrap iterations
```

**Fig 5. R Output 4.**

depending on the number of new cases at each age, which can be performed with functions `lyl_range` and `summary` with the appropriate `weights`. Age-specific estimates are available in Table 1. The weighted average for remaining life expectancy after the diagnosis is 34.0 years, which means that on average persons with the disease live an additional 34.0 years after the diagnosis. Alternatively, individuals with the disease lose 22.4 years of life after the diagnosis when considering a theoretical maximum of 95 years.

```
LYL <- lyl_range(data = diseased, t0 = age_disease,
t = age_death, status = cause_death, age_begin = 0,
age_end = 94, tau = 95)

LYL_ci <- lyl_ci(LYL, niter = 1000)

summary(LYL_ci, weights = diseased$age_disease)
```

The R output is shown in Fig 9.

Analogously, it is possible to estimate Life Years Lost over a range of ages for the general population, and summarize using the same weights as the population with the disease. The weighted average of remaining life expectancy for the general population for ages corresponding to the age-at-onset distribution is 40.2 years. The excess LYL among persons with a given disease after disease onset compared to the general population of same age are therefore 6.2 years (22.4 – 16.2), which can be decomposed into 4.3 years (19.3 – 15.0) due to natural causes and 1.9 years (3.1 – 1.3; with different results due to rounding error) due to unnatural causes.

```
LYL_ref <- lyl_range(data = simu_data, t = age_death,
status = cause_death, age_begin = 0, age_end = 94, tau = 95)

summary(LYL_ref, weights = diseased$age_disease)

lyl_diff(LYL_ci, LYL_ref, weights = diseased$age_disease)
```

```
summary(LYL45_ref)

## Estimates at age 45 years [maximum age tau = 95 years]
##
## ========================  ========  =======  ========
## \                         estimate  CI_left  CI_right
## ========================  ========  =======  ========
## Remaining life expectancy    32.31        -         -
## Total Life Years Lost        17.69        -         -
## - Due to Natural             16.71        -         -
## - Due to Unnatural            0.98        -         -
## ========================  ========  =======  ========
## *Confidence intervals can be estimated with function 'lyl_ci'
```

**Fig 6. R Output 5.**

```
lyl_diff(LYL45_ci, LYL45_ref)

## Estimates at age 45 years [maximum age tau = 95 years]
##
## =========================  ========  =======  ========
## \                          estimate  CI_left  CI_right
## =========================  ========  =======  ========
## Remaining life expectancy     -6.21    -6.37     -6.06
## Total Life Years Lost          6.21     6.06      6.37
## - Due to Natural               5.04     4.88      5.20
## - Due to Unnatural             1.17     1.06      1.29
## =========================  ========  =======  ========
## *95% confidence intervals based on 1000 bootstrap iterations
```

**Fig 7. R Output 6.**

The R output is shown in Fig 10.

When going from several age-specific LYL to one single estimate, we simply average over the observed distribution of onset ages. This information is usually available when collecting data from a group of patients with a disease. Alternatively, if information on the entire population (and not only the diseased) is available, one could use the distribution of onset ages conditional on disease occurrence estimated through transition intensities in the illness-death model [2]. It is also important to take into consideration the age-of-onset distribution in the population of interest, as ages with more cases will have larger weights in the overall estimate. Naturally, life lost will be larger at younger ages, simply because the potential of life lost at younger ages is larger than at older ages. Two diseases with the exact age-specific excess LYL could have different overall LYL if the age-of-onset distribution for the two diseases is different. The function summary without the appropriate weights will provide a table with the LYL at each specific age, which could be useful to investigate age-specific life lost.

### Step 5 Comparison to the general population when individual-level data are not available

In some situations, individual data from the general population might not be available. Fortunately, standard life tables are usually available from the Central Bureau of Statistics in each country or from the World Health Organization, and estimates from the population with a disease can be compared to these standard life tables. Standard life tables for Danish women in the period 2017-2018 (www.statistikbanken.dk) are provided in the 'pop_ref' dataset (available through the lillies package) to be used as example.

```
data(pop_ref)

head(pop_ref); tail(pop_ref)
```

The R output is shown in Fig 11.

For each age, the proportion of women alive and the mortality rate are provided. The estimation of excess LYL using life tables can be performed with the function lyl_diff_ref, and only one of the two age-specific measures provided (mortality rates or survival) is

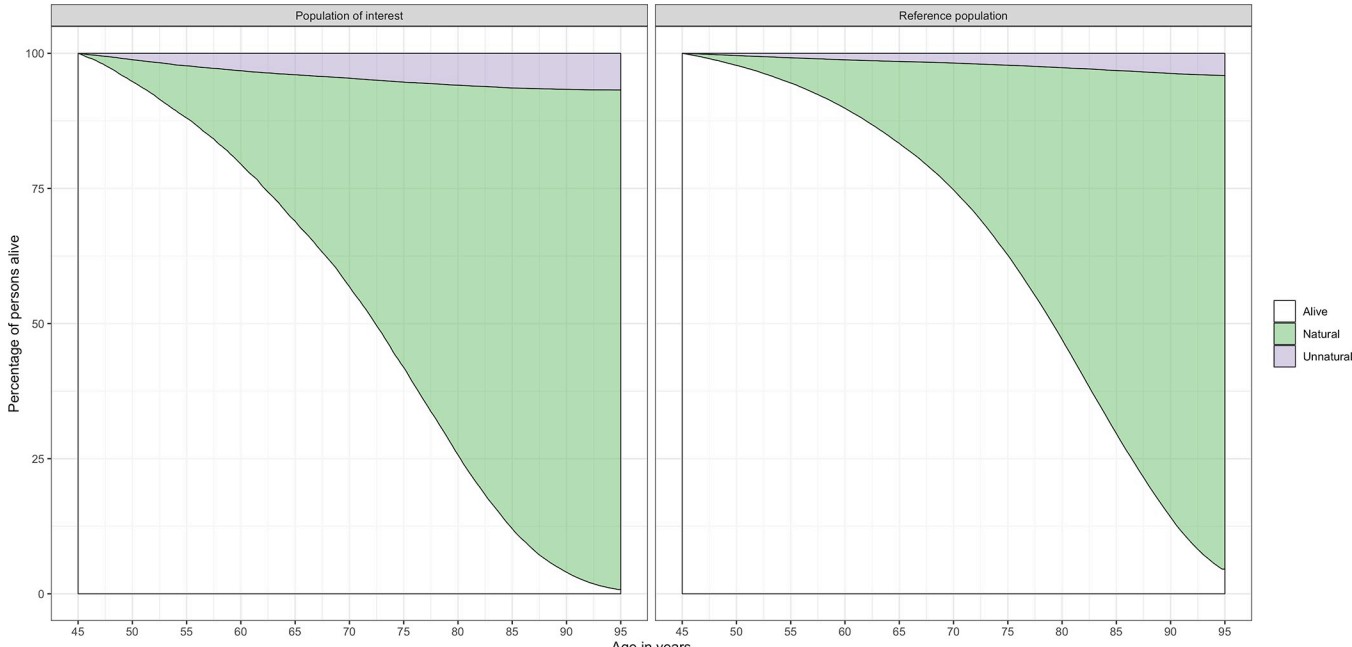

**Fig 8.** Survival curves and stacked cause-specific cumulative incidences for natural and unnatural causes of deaths for persons with a diagnosis of the disease (left panel) and the general population (right panel) alive at age 45 years. Details on how to interpret these figures are available in S1 Appendix.

sufficient. This function identifies whether the results for individuals with a disease is at one specific age or over a range of ages, or whether it includes confidence intervals, and returns the corresponding comparison.

```
lyl_diff_ref(LYL45, data_ref = pop_ref, age = age,
surv = survival)

lyl_diff_ref(LYL_ci, data_ref = pop_ref, age = age,
rates = mortality_rates, weights = diseased$age_disease)
```

The R output is shown in Fig 12.

If the simulated population used in this example were Danish women in 2017-2018, then the excess LYL at age 45 years could be estimated as 12.3 years. The average excess LYL among persons with a given disease after disease onset compared to the general population of Danish women of same age are 11.8 years (95% CI: 11.7 – 12.0).

## Excess Life Years Lost using aggregated-level data

We saw in the previous section that publicly available life tables can be used to compare life expectancy for a group of individuals with a disease with that of the general population. In some instances, individual-level data is not available for the group of individuals with the disease of interest either. However, it is still possible to estimate LYLs if the researcher has access to (i) number of new cases at each specific age, and (ii) age-specific mortality rates among those with the disease (or age-specific survival probability). A simulated dataset 'aggreg_data' is available through the package lillies as example for a disease with possible onset after

age 40 years. Note also that data is available only until age 90 years (the maximum age $\tau$ has to be set at 90 years for this example).

```
data(aggreg_data)

head(aggreg_data); tail(aggreg_data)
```

The R output is shown in Fig 13.

Excess LYL at one specific age using aggregated data can be estimated with function `lyl_aggregated`, providing aggregated data for those with the disease and also for the general population. A person with disease onset at age 70 years experiences an excess LYL of 7.7 years (12.6 years vs. 4.9 years in the reference population consisting of all Danish women, as in the previous example).

```
lyl_summary_data70 <-

    lyl_aggregated(data = aggreg_data, age = age,
rates = rate,

            data0 = pop_ref, age0 = age, surv0 = survival,

            age_specific = 70, tau = 90)

summary(lyl_summary_data70)

plot(lyl_summary_data70)
```

The R output is shown in Fig 14 and the plot generated is shown in Fig 15. On average, we can see that, for those alive at age 70 years, mortality after this age is much higher for a person with the disease of interest than from the general population.

Excess LYL averaged over the observed age-at-onset distribution can be estimated with function `lyl_aggregated_range`, providing the number of new cases at each age (`weights`).

```
lyl_summary_data <-

    lyl_aggregated_range(data = aggreg_data, age = age,

                rates = rate, weights = new_cases,

                data0 = pop_ref, age0 = age, surv0 = survival,

                age_begin = 40, age_end = 89, tau = 90)

summary(lyl_summary_data)
```

**Table 1. For each age *i* from 0 to 94 years: number of persons diagnosed with the disease ($n_i$); 95-restricted remaining life expectancy at each age for those with the disease ($LE_i^1$) and those from the general population ($LE_i^0$); total, natural and unnatural years of life lost at each age (denoted respectively as $_{95-i}\Theta_i$, $_{95-i}\Theta_i^1$ and $_{95-i}\Theta_i^2$ in S1 Appendix) for those with the disease ($LL_i^1$, $LLn_i^1$ and $LLu_i^1$, respectively) and those from the general population ($LL_i^0$, $LLn_i^0$ and $LLu_i^0$, respectively); and differences between these estimates ($LE_i^0 - LE_i^1$ or $LL_i^1 - LL_i^0$ for overall differences, and $LLn_i^1 - LLn_i^0$ and $LLu_i^1 - LLu_i^0$, respectively, for cause-specific differences). Weighted means are averages of each column weighted by the number of cases at each age, corresponding to means over the whole group of women with the disease. All columns are in years except 'Cases'.**

| Age *i* | DISEASED | | | | | GENERAL POPULATION | | | | DIFFERENCE | | |
|---|---|---|---|---|---|---|---|---|---|---|---|---|
| | Cases $n_i$ | Life exp $LE_i^1$ | Life lost $LL_i^1$ | Natur $LLn_i^1$ | Unnat $LLu_i^1$ | Life exp $LE_i^0$ | Life lost $LL_i^0$ | Natur $LLn_i^0$ | Unnat $LLu_i^0$ | Total | Natur | Unnat |
| 0 | 0 | 67.4 | 27.6 | 22.5 | 5.1 | 75.4 | 19.6 | 17.6 | 2.0 | 8.0 | 4.9 | 3.1 |
| 1 | 35 | 66.4 | 27.6 | 22.5 | 5.1 | 74.6 | 19.4 | 17.4 | 2.0 | 8.2 | 5.1 | 3.1 |
| 2 | 76 | 65.4 | 27.6 | 22.5 | 5.1 | 73.6 | 19.4 | 17.4 | 2.0 | 8.2 | 5.1 | 3.1 |
| … | … | … | … | … | … | … | … | … | … | … | … | … |
| 44 | 299 | 26.9 | 24.1 | 21.8 | 2.3 | 33.2 | 17.8 | 16.8 | 1.0 | 6.3 | 5.0 | 1.3 |
| 45 | 302 | 26.1 | 23.9 | 21.7 | 2.2 | 32.3 | 17.7 | 16.7 | 1.0 | 6.2 | 5.0 | 1.2 |
| 46 | 301 | 25.3 | 23.7 | 21.6 | 2.1 | 31.4 | 17.6 | 16.6 | 1.0 | 6.1 | 5.0 | 1.1 |
| … | … | … | … | … | … | … | … | … | … | … | … | … |
| 93 | 85 | 1.5 | 0.5 | 0.5 | 0.01 | 1.6 | 0.4 | 0.4 | 0.01 | 0.1 | 0.1 | 0.00 |
| 94 | 57 | 0.8 | 0.2 | 0.2 | 0.01 | 0.9 | 0.1 | 0.1 | 0.00 | 0.1 | 0.1 | 0.00 |
| Weighted means | | 34.0 | 22.4 | 19.3 | 3.1 | 40.2 | 16.2 | 15.0 | 1.2 | 6.2 | 4.3 | 1.9 |

The R output is shown in Fig 16. Persons with a diagnosis experience a remaining life expectancy after disease onset 6.2 years shorter than the reference population.

## Conclusions

The Life Years Lost is an informative measure of mortality associated with a given trait or disorder that has greater construct validity because it uses the observed age-at-onset. One of its main advantages is that it allows the decomposition of excess mortality into specific causes of death, which is important in order to examine the magnitude of each cause, and implement focused public health programs. In this paper, we present the `lillies` R package, which can be used to estimate Life Years Lost and we show how to implement the method using a simulated population and life tables (available through the package). With this package, epidemiologists, applied biostatisticians, and other researchers can easily estimate cause-specific Life

```
summary(LYL_ci, weights = diseased$age_disease)

## Estimates at ages 0-94 years [maximum age tau = 95 years]
##
## =========================  =======  =======  ========
## \                          estimate  CI_left  CI_right
## =========================  =======  =======  ========
## Remaining life expectancy    34.03    33.89     34.17
## Total Life Years Lost        22.40    22.26     22.53
## - Due to Natural             19.26    19.11     19.39
## - Due to Unnatural            3.14     3.00      3.26
## =========================  =======  =======  ========
## *95% confidence intervals based on 1000 bootstrap iterations
```

**Fig 9. R Output 7.**

```
summary(LYL_ref, weights = diseased$age_disease)

## Estimates at ages 0-94 years [maximum age tau = 95 years]
##
## ========================  ========  =======  ========
## \                          estimate  CI_left  CI_right
## ========================  ========  =======  ========
## Remaining life expectancy    40.20       -         -
## Total Life Years Lost        16.22       -         -
## - Due to Natural             14.95       -         -
## - Due to Unnatural            1.28       -         -
## ========================  ========  =======  ========
## *Confidence intervals can be estimated with function 'lyl_ci'

lyl_diff(LYL_ci, LYL_ref, weights = diseased$age_disease)

## Estimates at ages 0-94 years [maximum age tau = 95 years]
##
## ========================  ========  =======  ========
## \                          estimate  CI_left  CI_right
## ========================  ========  =======  ========
## Remaining life expectancy    -6.18    -6.31     -6.03
## Total Life Years Lost         6.18     6.03      6.31
## - Due to Natural              4.31     4.16      4.45
## - Due to Unnatural            1.87     1.72      1.98
## ========================  ========  =======  ========
## *95% confidence intervals based on 1000 bootstrap iterations
```

**Fig 10. R Output 8.**

Years Lost using their own data. The LYL estimates are meant to be used merely as a descriptive tool rather than one on which causal conclusions should be drawn. This is first of all because the question raised concerning life-years lost with a given disease does not correspond

```
head(pop_ref); tail(pop_ref)

##   age survival mortality_rates
## 1   0  1.00000         0.00295
## 2   1  0.99705         0.00017
## 3   2  0.99688         0.00010
## 4   3  0.99678         0.00007
## 5   4  0.99671         0.00004
## 6   5  0.99668         0.00007

##     age survival mortality_rates
## 95   94  0.15005         0.21685
## 96   95  0.11752         0.24071
## 97   96  0.08923         0.25943
## 98   97  0.06608         0.29024
## 99   98  0.04690         0.30181
## 100  99  0.03275         0.34200
```

**Fig 11. R Output 9.**

```
lyl_diff_ref(LYL45, data_ref = pop_ref, age = age, surv = survival)

## Differences in estimates comparing 'LYL45' with 'pop_ref'.
## Estimates at age 45 years [maximum age tau = 95 years]
##
## ========================  ========  =======  ========
## \                         estimate  CI_left  CI_right
## ========================  ========  =======  ========
## Remaining life expectancy   -12.31        -         -
## Total Life Years Lost        12.31        -         -
## ========================  ========  =======  ========
## *Confidence intervals can be estimated with function 'lyl_ci'

lyl_diff_ref(LYL_ci, data_ref = pop_ref, age = age, rates = mortality_rates, weigh
ts = diseased$age_disease)

## Differences in estimates comparing 'LYL_ci' with 'pop_ref'.
## Estimates at ages 0-94 years [maximum age tau = 95 years]
##
## ========================  ========  =======  ========
## \                         estimate  CI_left  CI_right
## ========================  ========  =======  ========
## Remaining life expectancy   -11.82   -11.96    -11.68
## Total Life Years Lost        11.82    11.68     11.96
## ========================  ========  =======  ========
## *95% confidence intervals based on 1000 bootstrap iterations
```

**Fig 12. R Output 10.**

to a well-defined intervention for which a randomized study could be conceptualized [19].
Further details about the method are available in S2 Appendix. For example, life expectancy is
estimated non-parametrically and, in some instances, a small number of individuals at risk
(especially at older ages) can lead to unreliable estimates. We provide a function to examine

```
head(aggreg_data); tail(aggreg_data)

##   age new_cases deaths       rate
## 1  40       851     37 0.02061281
## 2  41       798     48 0.02285714
## 3  42       837     48 0.02032176
## 4  43       815     51 0.01932550
## 5  44       791     84 0.02874743
## 6  45       828     69 0.02125693

##    age new_cases deaths      rate
## 45  84      5139   3932 0.1982155
## 46  85      4672   4154 0.2125787
## 47  86      4459   4112 0.2206245
## 48  87      4167   4134 0.2362961
## 49  88      3520   3975 0.2478334
## 50  89      3012   3783 0.2658842
```

**Fig 13. R Output 11.**

```
summary(lyl_summary_data70)

## Differences in estimates comparing 'aggreg_data' with 'pop_ref'.
## Estimates at age 70 years [maximum age tau = 90 years]
##
## ========================  ===========  =======  ==========
## \                          aggreg_data  pop_ref  Difference
## ========================  ===========  =======  ==========
## Remaining life expectancy         7.35    15.06       -7.71
## Total Life Years Lost            12.65     4.94        7.71
## ========================  ===========  =======  ==========
```

**Fig 14. R Output 12.**

whether there are enough observations in the population to obtain valid results. As in standard survival analysis, the LYL method works well with left truncation and right censoring as long as the assumption of independent censoring is met (i.e. those being censored at one specific time should be representative of those still at risk at that time). Additionally, when there is administrative censoring at one specific age (i.e. there are no observations by design after a certain age), as it was the case with the examples provided using individual-level data (95 years) and aggregated data (90 years), the LYLs can only be interpreted as life-lost before that specific age, included in the models as $\tau$. We also show in S2 Appendix how to examine whether the number of bootstrap iterations used to estimate confidence intervals are sufficient. Additionally, we show in S2 Appendix how to interpret negative excess LYLs; in a recent study [14], we observed that men with mental disorder had negative excess LYL related to cancer (i.e. the general population experienced a larger amount of life lost due to cancer than those with

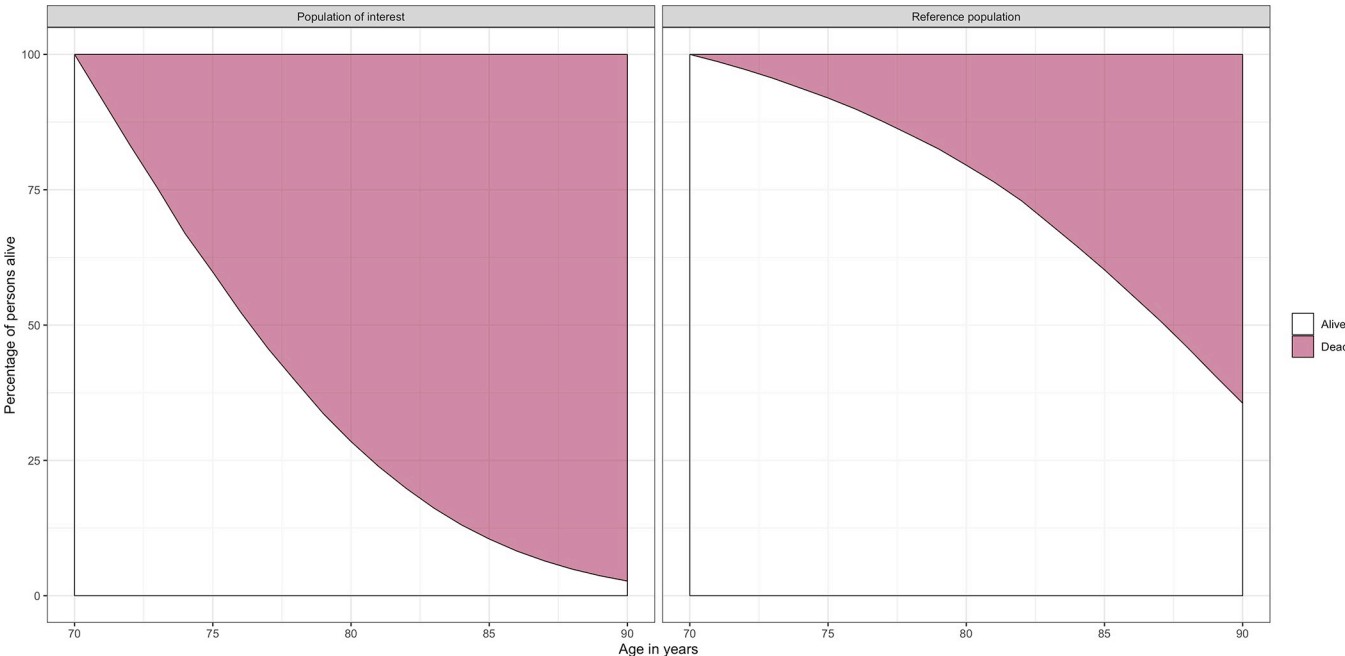

**Fig 15.** Survival curve and stacked cumulative incidence for mortality for persons with a diagnosis of the disease (left panel) and the general population (right panel) alive at age 70 years. Details on how to interpret these figures are available in S1 Appendix.

```
summary(lyl_summary_data)

## Differences in estimates comparing 'aggreg_data' with 'pop_ref'.
## Estimates at ages 40-89 years [maximum age tau = 90 years]
##
## =========================  ==========  =======  ==========
## \                          aggreg_data  pop_ref  Difference
## =========================  ==========  =======  ==========
## Remaining life expectancy        6.80    12.97       -6.17
## Total Life Years Lost            9.77     3.60        6.17
## =========================  ==========  =======  ==========
```

**Fig 16. R Output 13.**

mental disorders), an underappreciated feature in previous studies. This finding does not implicate that those with mental disorders have lower mortality due to cancer, but it relates to the ability of the LYLs to accommodate different causes of death. While men with mental disorders have higher rates of dying from cancer, they have even higher rates of dying of non-cancer causes of death, which precludes them of dying from cancer [14]. Finally, we provide the R code used to replicate the examples provided (S3 Appendix).

## Supporting information

**S1 Appendix. Basic concepts of survival analysis.**
(PDF)

**S2 Appendix. Technical extensions of the Life Years Lost method.**
(PDF)

**S3 Appendix. R code to replicate the examples.**
(R)

## Author Contributions

**Conceptualization:** Oleguer Plana-Ripoll.

**Funding acquisition:** Oleguer Plana-Ripoll, John J. McGrath.

**Methodology:** Oleguer Plana-Ripoll, Vladimir Canudas-Romo, Per Kragh Andersen.

**Software:** Oleguer Plana-Ripoll.

**Supervision:** Per Kragh Andersen.

**Validation:** Nanna Weye.

**Writing – original draft:** Oleguer Plana-Ripoll.

**Writing – review & editing:** Oleguer Plana-Ripoll, Vladimir Canudas-Romo, Nanna Weye, Thomas M. Laursen, John J. McGrath, Per Kragh Andersen.

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
