## [Decision Letter · Decision Letter 0]

22 Nov 2019

PONE-D-19-29790

lillies: an R package for the estimation of excess Life Years Lost among patients with a given disease or condition

PLOS ONE

Dear Plana-Ripoll,

Thank you for submitting your manuscript to PLOS ONE. After careful consideration, we feel that it has merit but does not fully meet PLOS ONE’s publication criteria as it currently stands. Therefore, we invite you to submit a revised version of the manuscript that addresses the points raised during the review process.

ACADEMIC EDITOR: 

As you can see from the reviewers comments (below) we feel this is a good description of the R package lillies and how to use it. The paper would need to be improved further by describing the underlying assumptions and limitations. For e.g. you mention that it is important to use a tau as large as possible so that few individuals are (right) censored. Indeed in most epidemiological research there is alot of right censoring. Would that disqualify this method? Is there a target value (proportion of right censoring) where you consider the method (in)appropriate? Specifically for competeing risks this is a cruicial problem and needs a thorough discussion. Further the method needs to be put in a causal framework of thinking (counterfactual comparison(s) identified by your estimator) as suggested by the reviewers. Also for your overall estimate the package (or at least your description) lack a comprehensive description of the underlying distribution of age at onset of the disease in the population/cohort. 12 lifeyears lost from birth is very different than 12 years lost from age 75 both in terms of interpretation and percentage etc. Please also adress all the specific comments raised by the reviweres. We are looking forward to your resubmission.

We would appreciate receiving your revised manuscript by Jan 06 2020 11:59PM. To enhance the reproducibility of your results, we recommend that if applicable you deposit your laboratory protocols in protocols.io, where a protocol can be assigned its own identifier (DOI) such that it can be cited independently in the future. For instructions see: http://journals.plos.org/plosone/s/submission-guidelines#loc-laboratory-protocols

We look forward to receiving your revised manuscript.

Kind regards,

Louise Emilsson

Academic Editor

PLOS ONE

Journal Requirements:

1. Our internal editors have looked over your manuscript and determined that it is within the scope of our Digital Health Technology Call for Papers. This collection of papers is headed by a team of Guest Editors for PLOS ONE: Eun Kyoung Choe (University of Maryland, College Park), Chelsea Dobbins (University of Queensland), Sunghoon Ivan Lee (University of Massachusetts, Amherst), and Claudia Pagliari (University of Edinburgh).The Collection will encompass a diverse range of research articles on digital health technologies ranging from technology design to patient care and health systems management.  Additional information can be found on our announcement page: https://collections.plos.org/s/digital-health-tech.

If you would like your manuscript to be considered for this collection, please let us know in your cover letter and we will ensure that your paper is treated as if you were responding to this call. If you would prefer to remove your manuscript from collection consideration, please specify this in the cover letter.

Reviewers' comments:

Reviewer's Responses to Questions

**Comments to the Author**

1. Is the manuscript technically sound, and do the data support the conclusions?

Reviewer #1: Yes

Reviewer #2: Yes

2. Has the statistical analysis been performed appropriately and rigorously? 

Reviewer #1: Yes

Reviewer #2: Yes

3. Have the authors made all data underlying the findings in their manuscript fully available?

Reviewer #1: Yes

Reviewer #2: Yes

4. Is the manuscript presented in an intelligible fashion and written in standard English?

Reviewer #1: Yes

Reviewer #2: Yes

5. Review Comments to the Author

Reviewer #1: The manuscript "lillies: an R package for the estimation of excess Life Years Lost among patients with a given disease or condition" is a very well written work that introduces the reader to the method of Life Years Losts in a very friendly, clear and comprehensive tone. The paper first describes the method and guides the reader through a number of steps, where the theoretical content is well complemented by the provided code and the interpretation of the output.

The paper does a great job by presenting the different features of the R package on different scenarios (like competing risks, aggregated-level data and comparison to general population using available life tables). The functions are very intuitive and the names of the arguments for each function are easy to remember given the provided examples. In addition, it is very considerate that the plots can be modified using ggplot2 arguments, and that the package provides handy functions to assess how small numbers can influence the estimates and how many iterations are necessary for bootstrap confidence intervals. The consistency in the coding style (for example using snake_case for variable names) is also appreciated.

Mayor comments:

1. Given that the work and the R package are targeted to applied researchers, it would be useful to emphasize under which underlying assumptions the estimates are valid in the Methods Section.

2. In addition, it would be useful to add in the Conclusions section, about the limitations of the method; particularly about providing causal interpretations based on the results.

Minor comments:

1. In the conclusions section, it would be useful to mention if there are other available R packages for the same method, and if so, compare the proposed package to the packages available.

2. At the end of Step 3, could you provide a reference after the sentence: “Although it might seem problematic to include persons with a disease in both the diseased and reference groups, this is analogous to standardized mortality ratios, which compare mortality in a group of persons to the one in the general population”. (Lines 305, 306, page 11)

Reviewer #2: This is a very well-written paper that addresses an interesting question – how to estimate average excess life years lost among individuals with a disease compared with the general population. The authors have prepared an R package to disseminate their work, facilitating the implementation of their ideas by a general audience. However, I have a couple of concerns:

1) The authors have not addressed or cited any of the important work that has been done in the field of causal inference in the last 30 years (e.g. any work by Jamie Robins, Tyler VanderWeele, Miguel Hernan). It appears that they are asking a fundamentally causal question: what are the average life years lost if everyone suffered from a particular disease compared with the “natural course” – the disease status they suffered in real life. This is a huge gap when thinking about this paper and should be addressed. The authors repeatedly describe causal quantities but do not consider any of the identifiability conditions for their claims, e.g. what counterfactual comparisons they are making by using their estimator. I believe this paper would be much stronger if it filled in this gap.

2) One issue which the authors do not consider is the fact that individuals may develop the disease at any time during the study. That brings into question the issue of having a clearly defined “time zero,” which can lead to a lot of immortal time bias (see Suissa 2007). The authors may want to familiarize themselves with the work by Danaei et al (2013) in the effect of statins on CVD, where they apply a “multiple trials” approach to estimating an average treatment effect (where your “treatment” would be disease versus no disease).

3) It seems that the authors choose to study a conditional survival quantity – probability of survival at time t if an individual survived to be 45 (figure 1, page 7). This should be more clearly specified and outlined. Also, there may be limitations to using a conditional survival quantity versus a marginal survival quantity.

6. PLOS authors have the option to publish the peer review history of their article (what does this mean?). If published, this will include your full peer review and any attached files.

Reviewer #1: No

Reviewer #2: No

---

## [Author Response · Author response to Decision Letter 0]

6 Dec 2019

Dear Dr Emilsson, 

Thanks for allowing us the opportunity to revise and resubmit this manuscript, and for providing us with many constructive suggestions. We have addressed each reviewer point below. Two versions of the revised manuscript have been uploaded (a ‘clean’ version, and one with track changes showing corrections). Additionally, we made changes to meet PLOS ONE’s style requirements (e.g. format of headings and subheadings), as suggested in the decision letter. Finally, thank you for alerting us to the interesting call for Digital Health Technology papers, but we do not think our manuscript is a good match, so we would like to have the manuscript removed from this collection consideration.

ACADEMIC EDITOR:

As you can see from the reviewers comments (below) we feel this is a good description of the R package lillies and how to use it. The paper would need to be improved further by describing the underlying assumptions and limitations. For e.g. you mention that it is important to use a tau as large as possible so that few individuals are (right) censored. Indeed in most epidemiological research there is alot of right censoring. Would that disqualify this method? Is there a target value (proportion of right censoring) where you consider the method (in)appropriate? Specifically for competeing risks this is a cruicial problem and needs a thorough discussion. Further the method needs to be put in a causal framework of thinking (counterfactual comparison(s) identified by your estimator) as suggested by the reviewers. Also for your overall estimate the package (or at least your description) lack a comprehensive description of the underlying distribution of age at onset of the disease in the population/cohort. 12 lifeyears lost from birth is very different than 12 years lost from age 75 both in terms of interpretation and percentage etc. Please also adress all the specific comments raised by the reviweres. We are looking forward to your resubmission.

RESPONSE: Thank you for constructive suggestions, which we address in this revision of the manuscript:

 We have now improved the manuscript by describing the assumptions and limitations. We agree there is usually a lot of right censoring in epidemiological research. The life-years lost method uses standard survival analysis techniques and can deal with right censoring without problems, as long as the assumption of independent censoring holds. A large amount of right censoring would lead to more uncertainty in the estimates, as in standard estimates such as the Kaplan-Meier or the Aalen-Johansen, but the point estimate will be unbiased if censoring is independent. We have now reworded some parts of the manuscript to include the assumption of right censoring. Specifically, we stated in the introduction that (new text underlined) “the LYL method also works with censoring at different ages, as in a standard time-to-event analysis (and therefore assuming independent censoring, i.e. those being censored at one specific time should be representative of those still at risk at that time)” (page 6, line 124); and in the conclusions that “As in standard survival analysis, the LYL method works well with left truncation and right censoring as long as the assumption of independent censoring is met (i.e. those being censored at one specific time should be representative of those still at risk at that time)” (page 19, line 549).

 Regarding the use of a tau as large as possible, we believe our explanations in the previous version of the manuscript were not clear. The choice of a tau is related to the availability of data after a certain age. When there is administrative censoring at one specific age tau (i.e. there are no observations after age tau), it is not possible to estimate the survival curve beyond that age, and consequently neither is the life expectancy. For this reason, the estimate obtained can only be interpreted as life expectancy (or life-years lost) before age tau. By choosing an age tau in which the survival curve is as low as possible (ideally zero), the life-years lost before age tau can be interpreted simply as the overall life-years lost (for example, life-years lost before age 120 years can be interpreted as overall life-years lost, as survival at age 120 years is zero). We have now rephrased the manuscript to make this statement clearer: “In brief, remaining life expectancy at age 45 years is estimated as the area under the conditional survival curve from age 45 years to ∞; however, this measure is sometimes ill-determined if there are censored observations and the curve does not reach zero (i.e. some persons are still alive at the end of the curve), as it is the case in this example (in fact, the survival curve cannot reach zero if the last person at risk is censored, even if this is the only censored observation in the dataset). One approach to overcome this limitation is the �-restricted mean lifetime, which can be interpreted as the average number of years lived before time �, and is defined as the area under the curve until time �: For this example, � has been set to 95 years, an age in which persons were censored if they had not died before. The estimate of LYL has therefore to be interpreted as life lost after the specific age (45 years in this example) and before age 95 years. Although the choice of 95 years is arbitrary, the life lost before � can be interpreted as total life lost if � is an age in which the survival probability is as low as possible (ideally zero). However, in other settings, the researchers might be interested in LYL before age 18 years for childhood disorders, or before retirement at age 68 years, for example” (page 7, line 171). We also added the following sentence to the conclusions: “Additionally, when there is administrative censoring at one specific age (i.e. there are no observations after a certain age by design), as it was the case with the examples provided using individual-level data (95 years) and aggregated data (90 years), the LYLs can only be interpreted as life-lost before that specific age, included in the models as �“ (page 19, line 551).

 The estimates obtained through the “lillies” package are valid even when the user aims to describe the reduction in life expectancy in people experiencing a disease or condition without making any causal claim. However, we agree it is important to not mislead the user about making any causal interpretation, and have added a paragraph on causal interpretations (page 18, line 539). See the comments to reviewer 2 for further details.

 Thank you for the comment about age-at-onset distribution, which we believe it is very important. We have now added the following section when describing the transformation from age-specific estimates to the overall one: “It is also important to take into consideration the age-of-onset distribution in the population of interest, as ages with more cases will have larger weights in the overall estimate. Naturally, life lost will be larger at younger ages, simply because the potential of life lost at younger ages is larger than at older ages. Two diseases with the exact age-specific excess LYL could have different overall LYL if the age-of-onset distribution for the two diseases is different. The function summary without the appropriate weights will provide a table with the LYL at each specific age, which could be useful to investigate age-specific life lost” (page 14, line 394).

REVIEWERS:

Reviewer #1:

The manuscript "lillies: an R package for the estimation of excess Life Years Lost among patients with a given disease or condition" is a very well written work that introduces the reader to the method of Life Years Losts in a very friendly, clear and comprehensive tone. The paper first describes the method and guides the reader through a number of steps, where the theoretical content is well complemented by the provided code and the interpretation of the output.

The paper does a great job by presenting the different features of the R package on different scenarios (like competing risks, aggregated-level data and comparison to general population using available life tables). The functions are very intuitive and the names of the arguments for each function are easy to remember given the provided examples. In addition, it is very considerate that the plots can be modified using ggplot2 arguments, and that the package provides handy functions to assess how small numbers can influence the estimates and how many iterations are necessary for bootstrap confidence intervals. The consistency in the coding style (for example using snake_case for variable names) is also appreciated.

 RESPONSE: Thank you for a nice comment.

Mayor comments:

1. Given that the work and the R package are targeted to applied researchers, it would be useful to emphasize under which underlying assumptions the estimates are valid in the Methods Section.

RESPONSE: The life-years lost method uses standard survival analysis estimates, e.g. Kaplan Meier for survival curves without competing risks, or Aalen-Johansen for cause-specific cumulative incidences. As such, the most important assumption is about independent censoring, i.e. those being censored at one specific time should be representative of those still at risk at that time. We have now included this assumption in the introduction of the method (page 6, line 124) and also in the conclusions section (page 19, line 549). See responses 1 and 2 to the Academic Editor suggestions for further details.

2. In addition, it would be useful to add in the Conclusions section, about the limitations of the method; particularly about providing causal interpretations based on the results.

RESPONSE: We have now included a paragraph about causal interpretations in the conclusions section (page 18, line 539). See the comments to reviewer 2 and response 3 to the Academic Editor suggestions for further details.

Minor comments:

1. In the conclusions section, it would be useful to mention if there are other available R packages for the same method, and if so, compare the proposed package to the packages available.

RESPONSE: As far as we know, there are no other packages available to estimate life-years lost averaged over the age-of-onset distribution. 

2. At the end of Step 3, could you provide a reference after the sentence: “Although it might seem problematic to include persons with a disease in both the diseased and reference groups, this is analogous to standardized mortality ratios, which compare mortality in a group of persons to the one in the general population”. (Lines 305, 306, page 11)

RESPONSE: Thank you, we have provided the following reference: Clayton D, Hills M. Statistical models in epidemiology. Oxford: Oxford University Press. 1993.

Reviewer #2:

This is a very well-written paper that addresses an interesting question – how to estimate average excess life years lost among individuals with a disease compared with the general population. The authors have prepared an R package to disseminate their work, facilitating the implementation of their ideas by a general audience. However, I have a couple of concerns:

 RESPONSE: Thank you for a thorough review and constructive suggestions.

1) The authors have not addressed or cited any of the important work that has been done in the field of causal inference in the last 30 years (e.g. any work by Jamie Robins, Tyler VanderWeele, Miguel Hernan). It appears that they are asking a fundamentally causal question: what are the average life years lost if everyone suffered from a particular disease compared with the “natural course” – the disease status they suffered in real life. This is a huge gap when thinking about this paper and should be addressed. The authors repeatedly describe causal quantities but do not consider any of the identifiability conditions for their claims, e.g. what counterfactual comparisons they are making by using their estimator. I believe this paper would be much stronger if it filled in this gap.

 RESPONSE: We agree that we have not mentioned any work in the field of causal inference, but we believe we are not making any causal claim in the entire manuscript. The “lillies” package is based on the life-years lost method, described by one of our co-authors previously (Andersen, 2017). This method allows to estimate the average reduction in life expectancy after disease onset experienced by those with a specific disease, but the method does not include any assumption that the reduction is actually caused by the disease. For example, we have recently used this method to show that men and women with mental disorders experience respectively 10 and 7 years shorter life expectancies after disease diagnosis compared to the general Danish population of same sex and age (Plana-Ripoll et al., 2019). However, we believe this is merely a descriptive estimate, as it is very difficult to fulfil the assumptions for causality (e.g. what would be the counterfactual of experiencing depression?). We think that making any causal interpretation of these estimates depend on each specific case, and the user of the package should be the one to take this into consideration. In any case, we agree it is important to make this point clear and we included a paragraph on causal interpretations in the conclusions section: “The LYL estimates are meant to be used merely as a descriptive tool rather than one on which causal conclusions should be drawn. This is first of all because the question raised concerning life-years lost with a given disease does not correspond to a well-defined intervention for which a randomized study could be conceptualized (Hernán and Robins, 2020).” (page 18, line 539). 

2) One issue which the authors do not consider is the fact that individuals may develop the disease at any time during the study. That brings into question the issue of having a clearly defined “time zero,” which can lead to a lot of immortal time bias (see Suissa 2007). The authors may want to familiarize themselves with the work by Danaei et al (2013) in the effect of statins on CVD, where they apply a “multiple trials” approach to estimating an average treatment effect (where your “treatment” would be disease versus no disease).

RESPONSE: We believe we are taking into consideration that individuals may develop the disease at any time during the study. In fact, we consider a variable of age at onset, which we treat as time-varying, and we consider individuals to experience the disease only since that age. In the dataset used as example, when estimating LYLs for those with a disease, we include an argument “t0 = age_disease” to specify that the diseased should enter the follow-up period only when they are diagnosed, and not since birth, precisely to avoid immortal time bias (page 8, line 203). In the first type of bias described in Suissa 2007 (misclassification of immortal time), individuals would be considered to have the disease since birth, which is exactly what we are recommending the users to avoid: “Note that the beginning of follow-up here is at age_disease to avoid immortal time bias (individuals survive until disease onset, therefore follow-up must start at disease onset, and not at birth, for the group of persons with the disease)” (page 8, line 203). The second type of bias described in Suissa 2007 (excluded immortal time) might seem what we are doing here, but in their case, there is a need for a “time 0” also among the unexposed because the time scale is time since exposure. However, in our analyses, age is always the underlying time scale, and individuals are followed since a pre-specified age, which is the same for exposed and unexposed. Consequently, we believe this is not creating immortal time bias.

3) It seems that the authors choose to study a conditional survival quantity – probability of survival at time t if an individual survived to be 45 (figure 1, page 7). This should be more clearly specified and outlined. Also, there may be limitations to using a conditional survival quantity versus a marginal survival quantity.

RESPONSE: The conditional survival curve is explained in Appendix S1. However, we have changed some text in the main manuscript to make it clearer it is the conditional survival curve we are estimating (page 7, lines 159, 164, 172). A main advantage of the life-years lost (compared to previous methods) is that it allows to take into consideration the observed age-at-onset of the disease, instead of assuming that all cases had onset at one particular age (e.g. at birth or at age 15 years). This, on the other hand, necessitates the use of a survival curve from disease onset and forward in time, i.e. a conditional survival curve, in order to avoid immortal time bias.

References

Andersen, P. K. (2017) ‘Life years lost among patients with a given disease’, Stat Med. 2017/06/07, 36(22), pp. 3573–3582. doi: 10.1002/sim.7357.

Hernán, M. A. and Robins, J. M. (2020) Causal Inference: What If. Boca Raton: Chapman & Hall/CRC.

Plana-Ripoll, O. et al. (2019) ‘A comprehensive analysis of mortality-related health metrics associated with mental disorders: a nationwide, register-based cohort study’, The Lancet. Elsevier, 394(10211), pp. 1827–1835. doi: 10.1016/S0140-6736(19)32316-5.

---

## [Decision Letter · Decision Letter 1]

8 Jan 2020

lillies: an R package for the estimation of excess Life Years Lost among patients with a given disease or condition

PONE-D-19-29790R1

Dear Dr. Plana-Ripoll,

We are pleased to inform you that your manuscript has been judged scientifically suitable for publication and will be formally accepted for publication once it complies with all outstanding technical requirements.

With kind regards,

Louise Emilsson

Academic Editor

PLOS ONE

Additional Editor Comments (optional):

Reviewers' comments:

Reviewer's Responses to Questions

**Comments to the Author**

1. If the authors have adequately addressed your comments raised in a previous round of review and you feel that this manuscript is now acceptable for publication, you may indicate that here to bypass the “Comments to the Author” section, enter your conflict of interest statement in the “Confidential to Editor” section, and submit your "Accept" recommendation.

Reviewer #1: All comments have been addressed

Reviewer #2: All comments have been addressed

2. Is the manuscript technically sound, and do the data support the conclusions?

Reviewer #1: Yes

Reviewer #2: Yes

3. Has the statistical analysis been performed appropriately and rigorously? 

Reviewer #1: Yes

Reviewer #2: Yes

4. Have the authors made all data underlying the findings in their manuscript fully available?

Reviewer #1: Yes

Reviewer #2: Yes

5. Is the manuscript presented in an intelligible fashion and written in standard English?

Reviewer #1: Yes

Reviewer #2: Yes

6. Review Comments to the Author

Reviewer #1: (No Response)

Reviewer #2: (No Response)

7. PLOS authors have the option to publish the peer review history of their article (what does this mean?). If published, this will include your full peer review and any attached files.

Reviewer #1: No

Reviewer #2: No

---

## [Editor Report · Acceptance letter]

21 Jan 2020

PONE-D-19-29790R1 

lillies: an R package for the estimation of excess Life Years Lost among patients with a given disease or condition 

Dear Dr. Plana-Ripoll:

I am pleased to inform you that your manuscript has been deemed suitable for publication in PLOS ONE. Congratulations! Your manuscript is now with our production department. 

With kind regards,

on behalf of

Dr. Louise Emilsson 

Academic Editor

PLOS ONE